# Understanding the Roles of the Hedgehog Signaling Pathway during T-Cell Lymphopoiesis and in T-Cell Acute Lymphoblastic Leukemia (T-ALL)

**DOI:** 10.3390/ijms24032962

**Published:** 2023-02-03

**Authors:** Alberto M. Martelli, Francesca Paganelli, Serena Truocchio, Carla Palumbo, Francesca Chiarini, James A. McCubrey

**Affiliations:** 1Department of Biomedical and Neuromotor Sciences, University of Bologna, 40126 Bologna, Italy; 2CNR-Institute of Molecular Genetics “Luigi Luca Cavalli-Sforza”, Unit of Bologna, 40136 Bologna, Italy; 3Department of Biomedical, Metabolic and Neuronal Sciences, University of Modena and Reggio Emilia, 41125 Modena, Italy; 4Department of Microbiology and Immunology, East Carolina University, Greenville, NC 27834, USA

**Keywords:** Hedgehog signaling, thymocyte differentiation, T-ALL, targeted therapy, crosstalk, signaling pathways

## Abstract

The Hedgehog (HH) signaling network is one of the main regulators of invertebrate and vertebrate embryonic development. Along with other networks, such as NOTCH and WNT, HH signaling specifies both the early patterning and the polarity events as well as the subsequent organ formation via the temporal and spatial regulation of cell proliferation and differentiation. However, aberrant activation of HH signaling has been identified in a broad range of malignant disorders, where it positively influences proliferation, survival, and therapeutic resistance of neoplastic cells. Inhibitors targeting the HH pathway have been tested in preclinical cancer models. The HH pathway is also overactive in other blood malignancies, including T-cell acute lymphoblastic leukemia (T-ALL). This review is intended to summarize our knowledge of the biological roles and pathophysiology of the HH pathway during normal T-cell lymphopoiesis and in T-ALL. In addition, we will discuss potential therapeutic strategies that might expand the clinical usefulness of drugs targeting the HH pathway in T-ALL.

## 1. Introduction

The Hedgehog (HH) signaling pathway was discovered more than 40 years ago in the fruit fly *Drosophila melanogaster*, where HH mutations disrupt the polarity of the larval segments [1]. The HH network is highly conserved from insects to mammals; this finding highlights its importance during the evolution of the species. This signaling cascade plays a major role during organism development by regulating cell fate, proliferation, survival, differentiation, and patterning [2,3]. However, over the last 15 years it has become apparent that aberrant activation of the HH signaling pathway is observed in many types of tumors, including medulloblastoma, rhabdomyosarcoma, basal cell carcinoma, glioblastoma, and cancers of the colon, lung, pancreas stomach, ovary, and breast [4]. Regarding hematological malignancies, deregulated HH signaling is implicated in the maintenance and expansion of leukemic stem cells (LSC), as well as in the development of drug resistance in acute myelogenous leukemia (AML) [5]. An HH pathway inhibitor, Glasdegib, has been approved in combination with low-dose cytarabine (ARA-C) for use in AML patients who are unfit for traditional intensive chemotherapy schemes [6]. T-ALL is a blood malignancy characterized by the rapid proliferation of lymphoblasts blocked at an early differentiation stage, thereby supplanting normal hematopoiesis. The cure rates for T-ALL patients reach 50% in adults and over 80% in children, thanks to current intensive chemotherapy protocols [7]. Nevertheless, primary resistance to treatment and relapse are observed in a significant number of patients, for whom prognosis remains poor, thereby underscoring the need for more efficient and targeted therapeutics [8]. HH signaling is also activated in lymphoid malignancies, including T-cell acute lymphoblastic leukemia (T-ALL) [9,10], where this signaling network is involved in cell proliferation, survival, and drug resistance [8]. Therefore, the HH signaling network could be a target for innovative therapies in T-ALL. Here, after a brief overview of HH signaling in healthy and cancer cells, we present a comprehensive and updated review of the relevance of the HH signaling network during T-cell lymphopoiesis and in T-ALL. We will also summarize novel strategies targeting the HH signaling network which could possibly lead to innovative therapies in T-ALL patients.

## 2. An Overview of Canonical HH Signaling Pathway 

The HH signaling pathway is a highly regulated, concerted network consisting of extracellular ligands, membrane receptors, cytoplasmic signaling molecules, co-regulators, and transcription factors. The interactions between components of this signaling cascade are spatially and temporally regulated to provide pathway activation only in the proper cell and tissue context. Typically, this signaling network is not active in mature cells, whereas its activation is necessary for tissue development and homeostasis [11]. HH signals play significant roles in the maintenance of pluripotent and somatic stem cell populations in the skin [12], nervous tissue [13,14], mammary gland [15], exocrine pancreas [16], as well as in prostate [17] and lung epithelia [18]. Accordingly, many of the target genes of this pathway control the expression of developmental regulators, including FGF4, Pax6-9, ABCG2, WNT, and HHIP [11]. HH signaling is also temporarily activated during tissue repair and wound healing, as stem cells are actively contributing to tissue regeneration [19]. 

The HH signaling network is extremely intricate due to the existence of canonical and non-canonical mechanisms of activation. In mammals, the activation of the canonical HH signaling pathway starts with the binding of one of three HH ligands, Sonic HH (SHH), Desert HH (DHH), or Indian HH (IHH), to a 12-pass transmembrane receptor, referred to as Patched (PTCH), which comprises two homologs, PTCH1 and PTCH2 [20]. Interestingly, all of the three ligands are post-translationally modified via the covalent attachment of palmitate and cholesterol moieties. These modifications are necessary for the proper secretion of the ligands and their extracellular distribution along the morphogenic gradients, as well as for maximal pathway activation [21]. 

In the absence of HH ligands, PTCH receptors localize to the primary cilium, where they prevent activation of the seven-pass transmembrane signaling protein Smoothened (SMO). SMO inhibition is achieved through the export of sterols from the inner leaflet of the plasma membrane, thereby depriving SMO of the sterols required for its activation [22]. Primary cilia detect extracellular cues and transduce them into cells, thereby regulating a variety of signaling pathways, including HH, WNT, NOTCH, and mitogen-activated protein kinase/extracellular signal-regulated kinase (MEK/ERK) [23]. For the scope of this review, it is important to emphasize that the primary cilium is detectable in 95–97% of bone marrow (BM) and peripheral blood mononuclear cells [24]. 

Upon binding of an HH ligand to a two-molecule PTCH complex, the N-terminal palmitate of the ligand is inserted into the sterol tunnel of one of the PTCH molecules, thereby blocking sterol export. Then, binding of a ligand to the second PTCH molecule induces endocytosis of the complex away from the primary cilium [25].

Once PTCH proteins leave the cilium, they lose their repressing activity on SMO. When SMO is inactive, Suppressor of Fused (SUFU) binds to the three GLI (from glioma-associated) zinc finger transcription factors (GLI1, GLI2, GLI3) and prevents their nuclear translocation and gene regulatory activities [26]. SUFU is a major repressor of GLI activity in unstimulated cells via direct binding through an N-terminal interacting site which sequesters GLI proteins in the cytoplasm [27]. Kinesin family protein 7 (KIF7) is a minor GLI protein repressor [28]. Moreover, several kinases (protein kinase A (PKA), casein kinase 1α (CK1α), and glycogen synthase kinase-3β (GSK-3β)) are recruited by SUFU/KIF7 and activated by PTCH. GLI phosphorylation by the aforementioned kinases marks them for proteolysis [29,30]. For instance, both GLI2 and GLI3 undergo limited proteolytical cleavage by Skp1-Cullin1-F-box protein (SCF) β-transducin repeat containing 1 (β-TrCP1) ubiquitin E3 ligase complex [31] and transform into their repressing forms (GLI2-R and GLI3-R). While GLI3 is most commonly present in its repressor form, GLI2-R is significantly less stable and is quickly degraded [32]. 

Upon HH ligand binding to PTCH, SMO accumulates in the primary cilium, thereby initiating the canonical HH signaling cascade [33,34,35,36,37]. SMO translocation to the primary cilium requires association with and phosphorylation at the C-terminus by casein kinase 1α (CK1α) and G Protein-Coupled Receptor kinase 2 (GRK2) [35,38,39]. Moreover, evidence suggests that the cyclic AMP (cAMP)-protein kinase (PKA) signaling pathway is also implicated in the lateral movement of SMO in mammalian cells [40]. Once SMO is translocated to the primary cilium, it inhibits GLI phosphorylation by PKA, CK1α, and GSK-3β, thereby sparing the transcription factors from proteolytic cleavage. Moreover, SMO relieves GLI sequestration from SUFU/KIF7 with the aid of Ellis–Van Creveld syndrome ciliary complex proteins, EvC and EvC2 [41,42,43]. GLI1, GLI2, and GLI3 are then licensed for gene transcriptional control in the nucleus [26,27,44] (Figure 1).

In addition to the canonical pathway of activation, GLI transcription factors could be activated via noncanonical mechanisms (which bypass the HH ligand–PTCH/SMO axis), mostly in cancer cells [45], as we shall discuss in Section 3.2.

## 3. Deregulated Canonical and Noncanonical HH Signaling in Cancer Cells

Initially, the HH pathway was linked to development anomalies [46,47,48]. However, a clear implication for deregulated HH signaling in human disorders was not well defined until 1996, when PTCH mutations were identified in basal cell nevus syndrome, also referred to as Gorlin–Goltz syndrome [49,50]. This rare syndrome is characterized by developmental anomalies which usually involve the skin and the skeleton [51]. Although basal cell nevus syndrome is associated with the development of tumors, such as basal cell carcinoma (BCC), medulloblastoma, and rhabdomyosarcoma [51], direct implication of the HH signaling pathway in cancer was demonstrated only later. 

Several lines of evidence suggest that aberrant activation of the HH signaling network drives cancer progression by regulating neoplastic cell proliferation and survival, metastasis, drug-resistance, and the expansion of cancer stem cells (CSCs) which are thought to be behind late-stage tumor recurrence [52,53]. In fact, GLI transcription factors control the expression of a plethora of genes involved in cancer cell proliferation and survival, epithelial-to-mesenchymal transition, drug resistance, and metastasis. These genes include *GLI1*, *PTCH1*, *PTCH2*, *SNAI1, HHIP1*, *MYCN*, *CCND1*, *CCND2*, *BCL2*, *CFLAR*, *FOXF1*, *FOXL1*, *FOXC2, TWIST2, PRDM1*, *JAG2*, *GREM1, ZEB1*, and *ZEB2* [54].

### 3.1. Canonical Activation

Four distinct mechanisms have been proposed to explain dysregulation of canonical HH signaling in neoplastic cells. However, these mechanisms are not necessarily mutually exclusive, which further complicates our understanding of the HH pathway in cancer [55]. 1. Type I ligand-independent: in this case, the pathway is constitutively upregulated independently from HH ligands. This is often caused by mutations in the *PTCH1*, *SMO*, and *SUFU* genes, as in the cases of BCC [56], rhabdomyosarcoma [57], and medulloblastoma [58]. 2. Type II ligand-dependent autocrine: this type of signaling occurs when the pathway is upregulated by HH ligands that are synthesized by the cancer cell itself. This modality of activation has been observed in a variety of cancers, including lung cancer, glioblastoma, pancreatic adenocarcinoma, and other tumors of the digestive tract [59,60,61,62]. 3. Type IIIa ligand-dependent paracrine: here, tumor cells produce ligands that induce HH signaling in other non-secreting malignant cells, as seen in the case of some subtypes of colorectal, pancreatic, endometrial, and ovarian cancers [63,64]. 4. Type IIIb ligand-dependent reverse paracrine: in this case, microenvironmental stromal cells secrete HH ligands, which then activate the pathway in cancer cells. This activation modality has been reported to occur in B-cell lymphomas [65], multiple myeloma [66], and some glioblastomas [67]. A schematic cartoon of the four modalities of canonical HH signaling activation is presented in Figure 2. 

### 3.2. Noncanonical Activation 

Noncanonical activation of the HH network, where PTCH/SMO are bypassed, is a typical driver in several malignant disorders [11]. The existence of noncanonical HH is extremely relevant in cancer cells, as it can partly explain constitutive or acquired resistance to SMO inhibitors. There are many ligand-independent noncanonical mechanisms of HH activation in tumor cells [45,68]. Noncanonical GLI signaling upregulation could depend on complex crosstalk between GLI proteins and other aberrantly activated signaling networks or oncogenic drivers, including RAS/RAF/MEK/ERK, PI3K/AKT/mTOR, dual-specificity tyrosine-(Y-) phosphorylation-regulated kinases (DYRKs), and Ewing sarcoma breakpoint region 1 (EWS)–Friend leukemia integration 1 (FLI1) [45,68]. Regarding the role of RAS/RAF/MEK/ERK in noncanonical HH signaling, it is known that ERK2 phosphorylates GLI1 at the N-terminus, thereby leading to its stabilization [69]. Moreover, ERK2-mediated phosphorylation weakens the binding of GLI1 to its negative regulator SUFU [70]. Both the events lead to upregulated GLI1 transcriptional activity. PI3K/AKT/mTOR signaling increases GLI protein activation and oncogenic function via at least two distinct mechanisms. Akt phosphorylates and inactivates GSK-3β which, when active, is a negative regulator of GLI1, most likely via its proteasomal degradation [71]. Hence, Akt overactivation stabilizes GLI1 [71]. mTOR upregulates p70S6K, which then phosphorylates GLI1 at Ser 84, thereby dissociating it from its repressor, SUFU. As a consequence, GLI1 migrates to the nucleus where it activates gene transcription [72]. As for DYRKs, while DYRK2 promotes the degradation of GLI2 [73], DYRK1B enhances GLI protein activity in an SMO-independent manner [74]. EWS–FLI1 is an oncogenic driver pathognomonic for Ewing sarcoma, which directly induces GLI1 expression [75].

Moreover, epigenetic regulation has been recently implicated in noncanonical HH signaling. For instance, while acetylation negatively modulates GLI1 and GLI2 transcriptional activity [76], deacetylation leads to enhanced GLI protein-dependent gene transcription [77]. Additionally, the epigenetic modifier bromo and extra C-terminal (BET) bromodomain 4 (BRD4) positively interacts with and upregulates the *GLI1* and *GLI2* promoters, thereby increasing gene transcription [78]. 

Examples of cancers displaying noncanonical HH signaling include, but are not limited to, esophageal carcinoma [72], glioblastoma [79], lung squamous cell carcinoma [80], Ewing sarcoma [75], medulloblastoma [81], and rhabdomyosarcoma [82]. However, in most of these tumors, aberrant canonical and noncanonical HH signaling coexist, thereby further complicating interpretation of the findings [68]. 

## 4. Therapeutic Targeting of HH in Cancer Cells 

The aberrant activation of HH signaling in cancer cells has prompted academic groups and pharmaceutical companies to develop several antagonists targeting the key components of this signaling pathway [83]. We will now briefly review the main compounds that have been employed to modulate the activity of HH signals in neoplastic cells.

### 4.1. SMO Inhibitors

The development of HH signaling pathway antagonists has primarily focused on targeting SMO. The first SMO inhibitor was cyclopamine, a natural occurring steroid alkaloid from the corn lily Veratrum californicum [84]. Several cyclopamine derivatives were then disclosed. Vismodegib (GDC-0449), a second generation cyclopamine derivative [85], was the first HH antagonist approved by the U.S. Food and Drug Administration (FDA) for treating BCC [86]. Sonidegib (NVP-LDE225) [87] is another FDA-approved SMO inhibitor used for BCC treatment [88]. PF-04449913 (Glasdegib) [89], has been approved for treatment of elderly AML patients in combination with low-dosage ARA-C [90]. Other SMO inhibitors include Saridegib (IPI-926) a semisynthetic SMO inhibitor, derived from a modification of the cyclopamine structure [91]; LY-2940680 (Taladegib) [92]; TAK-441 [93]; and BMS-833923 (XL139) [94]. All of these drugs have entered Phase I/II clinical trials for several types of cancer (Table 1). However, they have not been approved by the FDA. The efficacy of FDA-approved SMO inhibitors in BCC patients is limited, with a median duration response of about eight months and a median progression-free survival of approximately ten months [95,96]. Moreover, only 43% of advanced BCC and 30% of metastatic BCC patients are responsive to SMO inhibitors [97]. Furthermore, >20% of patients with advanced BCC who initially respond to vismodegib treatment later develop drug resistance, leading to tumor relapse [98]. As for AML, clinical activity of Glasdegib in combination with low-dose ARA-C was modest, regarding both overall survival (OS) (8.8 months vs. 5.5 months for the control cohort) and rate of complete remission (CR) (19.2% vs. 2.6%), as demonstrated by the Phase II Bright AML 1003 trial [99].

When Glasdegib was combined with intensive induction (intravenous ARA-C and daunorubicin) in untreated AML patients, CR rates were similar to historical controls, while there was a benefit in OS, with a median OS of 14.7 months in patients ≥ 55 years old vs. 8.7 months in historical controls. About 40% of patients were alive at 36 months [100]. In Table 1, we summarize the SMO inhibitors which have been used in clinical trials. 

### 4.2. Targeting GLI Transcription Factors 

This could be the most promising strategy for inhibiting HH signals, as therapeutics targeting GLI and inhibiting its DNA-binding activity may also blunt the noncanonical HH signaling route [101]. GLI-antagonist 58 and 61 (GANT 58 and GANT 61) [102] are small molecules that have been widely used in preclinical cancer models, where they display cytotoxic activity [103], induce apoptosis [104], and reduce NF-κB activity [105]. Moreover, GANT 61 is also effective in inhibiting the proliferation and self-renewal of CSCs [106,107]. Interestingly, GANT 61, when combined with the mTOR inhibitor rapamycin, is more efficient than cyclopamine in downregulating CSC-related properties (e.g., expression of the stem cell marker CD133, multicellular sphere formation) [108]. However, the clinical development of GANT 61 has been hampered by its poor pharmacokinetic profile; hence, GANT 61 has not entered clinical trials yet. Glabrescione B (GlaB) is a natural isoflavone, found in the seeds of Derris glabrescens, which binds the GLI1 zinc finger, thereby interfering with its interaction with DNA. GlaB inhibits the growth of HH-dependent cancer cells in vitro and in vivo, and inhibits the self-renewal activity and clonogenicity of cancer stem cells [109].

Arsenic trioxide (ATO) is capable of antagonizing the HH signaling pathway [110]. The mechanisms by which ATO downregulates GLI-dependent transcription are not clear yet. In the short term, ATO blocks HH-induced ciliary accumulation of GLI2, whereas after a prolonged incubation it decreases the steady-state levels of GLI2 [110]. However, these findings could not be confirmed by a different group, which reported that ATO directly binds to and inhibits GLI1 that, however, is not degraded. Moreover, ATO-induced inhibition of GLI1 activity does not require primary cilia [111]. It is worth recalling that ATO in combination with all-trans retinoic acid has completely redefined the treatment landscape for acute promyelocytic leukemia (APL) [112]. In APL, the main mechanism of action of ATO is through the proteasomal-dependent degradation of the promyelocytic leukemia-retinoic acid receptor-α (PML-RARα) fusion protein [113]. Interestingly, the HH pathway is activated in APL cells where ATO decreases the expression of some components of this signaling network, including GLI2, SMO, and PTCH [114]. The efficacy of ATO in combination with itraconazole has been evaluated in refractory metastatic BBC patients (NCT01791894); however, no clinical benefits were observed [115]. Itraconazole is a systemic antifungal drug which inhibits SMO by preventing its ciliary accumulation caused by HH ligands [116]. 

### 4.3. Targeting HH Signaling via Nanoparticles

It is becoming increasingly appreciated that nanoparticle-based drug delivery platforms have the potential for ameliorating the current anti-neoplastic treatments. For instance, nanoparticles offer several advantages including enhanced solubility, stability, and bioavailability of the transported drugs [117]. Nanocarriers could increase the efficacy of drugs by optimizing both pharmacokinetic and biochemical properties of the therapeutics. Hence, patients might endure higher doses of drugs while experiencing less severe side effects. In particular, nanoparticles carrying HH inhibitors could be useful for treating cancers that have metastasized and became resistant to free HH inhibitors [118]. Nanoparticles carrying natural compounds targeting HH signaling have also been successfully employed in preclinical cancer models [119]. Nevertheless, significant challenges remain to be overcome for a successful use of nanoparticles as drug carriers, including the preferential uptake of nanoparticles in the cells of the reticuloendothelial system as well as in the liver and the spleen, and the difficult penetration through the components of the extracellular matrix. 

## 5. HH Signaling and T-Cell Lymphopoiesis

Lymphopoiesis refers to the production of B-cells, T-cells, and natural killer (NK) cells. It comprises a series of maturational steps from hematopoietic stem cells (HSCs) to common lymphoid progenitors (CLPs), to fully differentiated B-, T-, and NK cells [120,121,122]. 

The relevance of HH signaling in adult HSCs is still controversial and ill-defined [123]. Some studies have demonstrated that it is dispensable [124,125], while others reached the conclusion that HH signaling is implicated in the expansion and differentiation of HSCs in both zebrafish [126] and mouse [127] models. While Gering at al. [126] took advantage of a loss-of-function SHH mutation, Merchant and coworkers [127] demonstrated that GLI1 is a regulator of self-renewal of HSCs and drives myeloid cell proliferation. The genetic ablation of GLI1 seems to impair hematopoiesis in situations of stress, after a cytotoxic injury, or in response to stimulatory cytokines. It might be that the wide range of results obtained simply reflects the different experimental models employed, the inherent complexity of the HH pathway, the promiscuous receptor–ligand interactions, and the possible crosstalk with other signaling cascades [128]. The fact that none of the therapeutics targeting HH signaling have been reported thus far to cause hematopoietic toxicity during clinical trials may support the idea that this pathway is not involved in adult normal hematopoiesis [123], although additional studies are certainly required. 

As to lymphopoiesis, the development of T- and B-cells is blocked in the BM at the level of CLPs if PTCH expression is abrogated in mice, whereas cells of myeloid lineage develop normally [129]. Another relevant finding of this work is that PTCH function is controlled by BM stromal cells, presumably via the secretion of HH ligands. Therefore, PTCH would act as an inducer of lymphoid versus myeloid lineage commitment [129], although this conclusion has not been confirmed by others [127].

Regarding T-cell lymphopoiesis, multiple lines of evidence indicate that HH signals are fundamental for T-cell development in both the embryonic and the adult thymus (reviewed in [130,131]). In the embryo, early progenitor T-cells enter the thymus through the capsule, then migrate towards the center as they mature [131]. In contrast, in the adult thymus, thymic seeding progenitors from the BM enter the thymus through blood vessels located at the corticomedullary junction. The T-cell progenitors then migrate across the cortex while undergoing an expansion and gene rearrangement phase. Subsequently, they initiate a reverse migratory path across the cortex toward the medulla, proceeding to final maturation [132,133]. 

The T-cell differentiation process comprises several phases, consisting of double negative (DN; CD4^-^CD8^-^), double positive (DP; CD4^+^CD8^+^), and, finally, single positive (SP; either CD4^+^ or CD8^+^) mature T-cell stages [134]. DN thymocytes may differentiate to DP cells via an intermediate CD8^+^ immature single positive (ISP) subpopulation [135]. Based on the expression status of CD25 and CD44, the DN stage could be further divided into four smaller cell subpopulations: DN1 thymocytes, which are not yet fully committed to the T-cell lineage, are CD25^-^CD44^+^, DN2 thymocytes are CD25^+^CD44^+^, DN3 thymocytes are CD25^+^CD44^-^, and DN4 thymocytes are CD25^-^CD44^-^ [136]. DN3 cells express a constitutively active pre-T-cell receptor (pre-TCR, which comprises the invariable pre-T α chain and the rearranged TCRβ chain). T-cell commitment towards the αβTCR positive lineage takes place during the later DN stages of maturation, at which γδTCR T-cells split off. DP thymocytes cease to proliferate and rearrange their TCRα loci. After undergoing both positive (to ensure appropriate MCH restriction [137]) and negative (to eradicate potentially self-reactive clones [138]) selection, thymocytes expressing functional αβTCRs commit to either CD4^+^ or CD8^+^ SP cells. Finally, SP CD4^+^ and CD8^+^ cells leave the thymus and populate the peripheral lymphoid tissue/organs [139,140,141]. 

The maturation of T-cells is strictly dependent on the interactions between the developing lymphocytes and the thymic stroma, which consists of thymic epithelial cells (TECs) and mesenchyme-derived cells [142]. Two populations of TECs have been identified: medullary (m) TECs and cortical (c) TECs, displaying different locations and functions [143,144] (Figure 3). The T-cell developmental program is orchestrated by the inputs of multiple signaling networks. For instance, early progenitor T-cells are induced to grow and proliferate mainly through interleukin-7 (IL-7) [145,146,147] and stem cell factor (SCF) [148]. The activation of NOTCH1 signaling in uncommitted lymphoid progenitors circulating in the thymic cortex is essential for lineage commitment toward a T-cell fate [149] and for proliferation of DN3 thymocytes [150]. The WNT signaling network is crucial for the transition from the CD8^+^ ISP stage to the CD4^+^CD8^+^ double positive stage [151,152]. In addition, PI3K signaling is critical for supporting CD8^+^ ISP maturation to CD4^+^CD8^+^ DP cells [153]. 

Components of HH signaling have been identified in both human [154] and mouse [155] thymi. TECs produce all the three HH ligands [154]. Fetal TECs also express the machinery to transduce HH signals, i.e., SMO, PTCH1, and GLI3 [131,156]. HH signaling plays important roles in the development of mTECs [157]. 

Regarding T-cell progenitors, IHH is expressed by thymocytes at the DP stage [158]. PTCH1 and SMO are expressed mainly by immature thymocytes. In particular, SMO displays a peak of expression at the DN1 and DN2 stages, while it is downregulated at subsequent stages, disappearing at DN4 [159]. GLI1 shows its highest expression in the DN2 and DN3 subsets, and then decreases to extremely low levels in DN4 and DP thymocytes. GLI2 is expressed at the highest levels at the DN1 and DN2 stages, downregulated at DN3, and then upregulated again at DN4 [160]. GLI3 is not expressed in adult mouse thymocytes; however, it shows a high level of expression in DN1 and DN4 fetal thymocytes [161]. 

There is a general agreement that HH signaling, through SHH (secreted by cTECs), SMO, GLI2, and GLI3, is an essential positive regulator of the early stages of thymocyte maturation in the fetal thymus, where it controls proliferation and survival of DN1 cells as well as their differentiation to DN2 thymocytes [131,159,162]. For instance, when Smo is selectively ablated at the DN1 stage, the thymus displays considerable atrophy characterized by a reduction of the DN and DP T-cell subsets due to increased apoptosis of DN2 and DN3 T-cells [159]. Mechanistically, Smo ablation results in downregulation of antiapoptotic Bcl2, Mcl1, and Ccnd2, whereas Cdkn1a, which encodes the cell cycle negative regulator p21^cip^, is increased [159]. An impaired differentiation to the DN2 subset has been also observed in Gli2^-/-^ or Gli3^-/-^ thymi [160,161], whereas Gli1 was shown to be dispensable for the DN1 to DN2 maturation, as GLI3 acts as a repressor of GLI1 in DN1 cells [163] (Figure 3). 

Regarding later stages of thymocyte differentiation, several lines of evidence indicate that HH signaling plays a negative role in the transition from DN3 to DN4 as well as from DN to SP subsets. Early experiments, performed in vitro using fetal thymus organ cultures (FTOCs) treated with recombinant SHH, demonstrated that thymocyte development was blocked at the DN3 stage [155]. Moreover, Smo deletion after pre-TCR development at the DN4 stage increased the DP and SP subsets [159]. Furthermore, both the E17.5 Gli2^-/-^ thymus and the E16.5 Gli3^-/-^ thymus display twice as many thymocytes as the control thymus [160,161]. This observation is consistent with the expression pattern of GLI2, which shows a threefold increase in DN4 in comparison with DN3 subpopulation. 

Although earlier studies have focused on the importance of SHH produced by TECs during early stages of T-cell development [155,162], a subsequent study suggests that IHH, while promoting T-cell development before pre-TCR signaling, is a negative regulator of T-cell maturation after pre-TCR signal transduction at the DN3 stage. Interestingly, IHH is abundantly produced by the DP population and negatively feeds back to the DN3 subset [158]. In fact, the high expression of IHH in the DP subset, which has the lowest expression of GLI1, implies that DP thymocytes do not respond themselves to HH signals but rather signal back to more immature thymocytes. Although IHH has a redundant effect at the DN1 to DN2 transition, as SHH could compensate for IHH absence, IHH has a unique effect on the negative regulation of the transition from DN3 to DP cells [158]. Overall, these findings indicate that IHH, GLI2, and GLI3 play negative roles in thymocyte expansion and differentiation to DP after pre-TCR signaling. 

Several studies have demonstrated that mTEC-secreted SHH also negatively regulates the transition from the DP to SP mature T-cells, most likely via a lowering of the TCR signaling strength. These investigations took advantage of several murine embryonic models (Shh^-/-^, Gli2^-/-^, Gli2A-transgenic, Gli2R-transgenic, Gli2ΔC-tramsgenic, Gli3^-/-^) [131,157,163,164,165,166]. Nevertheless, more recent evidence showed that GLI3, produced by TECs, represses SHH expression, thereby promoting positive selection from CD4^+^ CD8^+^ DP cells to CD4^+^ CD8^−^ SP lymphocytes [167]. The findings summarized above show how HH signaling may have opposite effects on thymocyte differentiation, similar to those reported in different organs/tissues, including the retina and gut [168,169]. 

Most of the studies on HH signaling in the thymus have been performed using animal models. However, a couple of studies have been accomplished in vitro using human cells. An early investigation reached the conclusion that SHH increases the survival of CD34^+^ thymocyte precursors while inhibiting their proliferation and maturation to DP cells [170]. These effects were mainly attributed to the capability of SHH to counteract the proliferation and differentiation of CD34^+^ progenitor cells induced by IL-7 signaling. In a subsequent study, FTOC experiments indicated that HH signaling attenuated the percentage of both CD4^+^ SP cells and CD8^+^SP cells, as well as increased the proportion of DP cells in thymi of children [157]. 

## 6. T-ALL

T-ALL comprises 10–15% of pediatric and 20–25% of adult cases of ALL and is considered to be a high-risk subtype of ALL [171]. In recent years, there has been an outstanding progress in T-ALL treatment, especially in pediatric patients, which is reflected by the increase in the 5-year OS, from about 50% in the 1970s to up to 80–85% in the most recent studies [172]. However, front-line therapy, based on chemotherapy and radiotherapy, fails in 15–20% of children and 50–70% of adults [173,174], and these patients face a very poor outcome [175,176]. T-ALL is a highly heterogeneous disease, whereby a wide spectrum of genetic anomalies [177] and microenvironmental factors [178] alter growth, proliferation, survival, and differentiation during T-cell development. Based on the maturational stage of healthy T-cell development at which leukemic transformation occurs, T-ALL can be subclassified into early T-lineage progenitor T-ALL (ETP-ALL, which displays a block at the earliest stages of T-cell differentiation [179]), early cortical, and late cortical or mature subtypes [180]. At present, however, the classification and stratification of T-ALL patients is preferentially based on the specific mutational profile, gene expression signatures, and genomic alterations of leukemic samples [181,182]. ETP-ALL (characterized by a very immature CD4^−^ CD8^−^ DN phenotype and the expression of both stem cell and myeloid-associated markers [183], commonly display mutations in transcription factors associated to hematopoietic and T-cell development (*GATA3*, *ETV6*, *RUNX1*), to epigenetic regulators (*DNMT3A*, IDH1, *IDH2*, *EZH2*,), and signaling pathways, including *FLT3* and NRAS [184]. Cortical T-ALLs (characterized by the expression of CD1a and a CD4^+^CD8^+^ DP phenotype), are typically associated with genetic alterations in *TLX1*, *TLX3*, *NKX.2.1*, and *NKX.2.2* homeobox genes, as well as in *CDKN2A*, *TAL1* and *LMO1*,2 [180,185]. *TAL1* and *LMO1*,2 expression are also associated with mature T-ALLs displaying surface CD3 [180,181]. *NOTCH1* oncogenic anomalies are frequent in cortical T-ALLs and can be detected in up to 40% of mature T-ALLs [181], whereas ETP T-ALLs display a lower prevalence of *NOTCH1* mutations [171]. Although several signaling pathways are aberrantly activated in T-ALL (reviewed in [8]), no targeted therapeutics have thus far been demonstrated to improve T-ALL prognosis [186]. In fact, at variance with kinase-driven leukemias such as chronic myeloid leukemia (CML) and Philadelphia-positive B-cell precursor acute lymphoblastic leukemia (B-ALL), T-ALL causes the aberrant expression of genes encoding transcription factors (e.g., TAL1, TAL2, LYL1, BHLHBL1, LMO, LMO2, TLX1/HOX11, HOX11L2, HOXA, etc., reviewed in [8]) which are notoriously very difficult to target [187]. Moreover, immunotherapy, such as chimeric antigen receptor T-cell (CAR-T) therapy, which has been remarkably successful in B-ALL patients [188], is still in its infancy for treatment of T-ALL [189], thereby highlighting the urgent need for novel targeted therapies for this hematological malignancy.

## 7. HH Signaling in T-ALL

Regarding hematological cancers, aberrant HH signaling has been mainly investigated in AML [190], where it displays leukemogenic potential [191]. The reasons for deregulated HH signaling in AML are far from clear; however, they could include increased paracrine DHH secretion by BM stromal cells [192], silencing of *GLI3* transcription via hypermethylation [193], expression of the *CBFA2T3-GLIS2* fusion gene which leads to upregulation of HH-related genes (*CRISP3*, *GATA3*, *H2AFY*, *NCAM1*) [194], and overexpression of *GLI1* [195]. Importantly, HH signaling is highly active in drug-resistant AML cells, whereas the activation is less pronounced in both drug-sensitive cells and non-refractory primary leukemic cells [196]. In November 2018, the U.S. FDA approved the SMO inhibitor, Glasdegib, in combination with low-dose ARA-C, for the therapy of newly diagnosed AML patients who are >75 years old or who have comorbidities that would preclude the use of intensive induction chemotherapy [197]. Deregulated HH signaling has also been identified also CML [198], multiple myeloma [199], chronic lymphocytic leukemia [200], and B-cell non-Hodgkin lymphomas [201]. 

The first reports suggesting a possible activation of the HH network in T-ALL cells were published in 2009 and 2011. They showed that the SMO inhibitor cyclopamine synergized with either quercetin [202] or a NOTCH inhibitor [203] in suppressing the growth of human T-ALL cell lines. 

### 7.1. Mechanisms of HH Signaling Activation in T-ALL

The causative factors contributing to HH signaling upregulation in T-ALL cells are far from being clear. The available evidence suggests that activation could depend on genetic aberrations, autocrine signaling, and ligand-independent noncanonical mechanisms.

#### 7.1.1. Activating Mutations

Mutations in critical components of the HH network have been identified in T-ALL patients. Dagklis and coworkers [9] discovered 9 previously unreported mutations in a cohort of 67 pediatric and adult T-ALL patients. Based on the analysis of the germline DNA, four of the mutations were identified as somatic mutations, two were present in germline DNA, while the other three mutations were identified in samples for which no germline DNA was available. Therefore, they could have represented true somatic mutations. Two of the somatic alterations were truncating SMO mutations, while the other two were point mutations of GLI1 (S538F) and GLI3 (G727R). Of note, the SMO mutations are located in the C-terminus of the protein, a domain containing several arginine clusters that are instrumental in blocking the cell surface expression of SMO and inhibiting its conformational switch to the active form [204]. On the other hand, the GLI mutations are both located downstream of the zinc finger domain and predicted to be damaging for protein function(s) [205]. Interestingly, the SMO, GLI1, and GLI3 mutations were still detectable at high frequency during disease remission, hence they could be de novo genetic anomalies acquired early in the life of the neoplastic hematopoietic cells [9]. 

In a subsequent study, 20 monoallelic anomalies of genes encoding components of the HH pathway were identified in 17 of 109 (16%) childhood T-ALL samples [206]. Two patients harbored monoallelic mutations of two HH pathway genes, either PTCH1/GLI2 or GLI1/SUFU, while a third case harbored two distinct mutations of GLI3. The remaining 14 T-ALL patients all had only a single mutation. PTCH1 was the single most commonly mutated gene, with nine patients harboring heterozygous missense mutations. However, there were also five GLI3, three GLI2, two GLI1, and one SUFU anomalies. Interestingly, the sequencing of remission specimens, which were available in five out of the nine cases with PTCH1 mutations, showed that three of the anomalies were absent at remission, thereby indicating a somatic origin, whereas two were detected also in the remission samples. However, due to the lack of non-hematopoietic germline specimens, it was impossible to distinguish whether these were germline mutations or somatic mutations associated with clonal malignant lymphopoiesis [206]. HH pathway genetic anomalies co-existed with some oncogenic lesions typical of T-ALL, including mutations of NOTCH1 and PRC2, as well as with deletions of CDKN2A and TAL1. This observation indicates that HH genetic alterations may cooperate with classical T-ALL oncogenic anomalies in T-cell transformation. It was also found that Jurkat T-ALL cells harbor a pathogenic PTCH1 mutation, encoding a p.Gly68fsX5 truncated PTCH1 protein which is unable to silence HH signaling. When Jurkat cells were transduced with wild type PTCH1, a downregulation of MYCN mRNA expression was observed. Interestingly, the analysis of MYCN RNA expression in primary T-ALL patient samples documented that HH pathway mutations were associated with an increased expression of MYCN. This finding indicates that MYCN is a transcriptional target of HH signaling in human T-ALL cells [206]. The incidence of HH pathway mutations was similar in the study by Dagklis and colleagues [9] and in that by Burns and coworkers [206]. In contrast, in a whole-genome sequencing (WGS) analysis, Liu and colleagues [207] identified HH pathway mutations in only 4 out of 264 T-ALL patients analyzed. A possible explanation for this apparent discrepancy is the association of HH signaling mutations with induction failure cases that were deliberately excluded from the WGS analysis, due to lack of samples from patients in remission which could allow one to distinguish between somatic and germline mutations.

#### 7.1.2. Autocrine Mechanism of HH Activation

Another mechanism of HH network activation in T-ALL could be the existence of an autocrine loop, as suggested by a study, based on RNA sequencing, in which 139 T-ALL samples were analyzed and compared with healthy T-cell progenitors [10]. An ectopic expression of SHH, IHH, and GLI1 was observed in 30 specimens (22%) and there was a robust correlation of GLI1 expression levels with the expression of the 2 HH ligands. Moreover, some GLI1 target genes (PTCH2, CCND1, BCL2) were upregulated in samples expressing high SHH/IHH/GLI1 levels. In contrast, the expression of SHH/IHH/GLI1 was much lower (or nonexistent in the case of SHH) in normal T-cell subsets [10]. These observations suggest, but do not prove, the existence of an autocrine HH activation mechanism in a subset of T-ALL patients.

#### 7.1.3. Noncanonical Mechanisms of HH Activation and Crosstalk with Other Networks

A few therapeutically relevant molecular pathways which crosstalk with HH signaling have been identified in T-ALL cells. Some of these crosstalks are indicative of the existence of noncanonical mechanisms of HH signaling upregulation. 

Two signaling pathways which are frequently overactive in T-ALL are the PI3K/Akt/mTOR and MEK/ERK networks [208,209]. Either perifosine (an Akt inhibitor, [210,211]) or the MEK inhibitor PD98059, decreased the expression levels of GLI1, thereby suggesting an involvement of the PI3K/Akt and MEK/ERK pathways in the regulation of noncanonical HH signaling [212]. These results are consistent with previous findings showing that these two pathways could upregulate GLI1/GLI2 expression levels or transcriptional activity in other cancer types, including glioblastoma, melanoma, colon cancer, and pancreatic adenocarcinoma [213,214,215,216]. The exact mechanisms by which PI3K/Akt/mTOR control GLI1 expression remains unclear. However, they could involve GSK-3β, as either pharmacological inhibition or genetic silencing of GSK-3β increased the expression of GLI1 [217,218]. These observations suggest that active GSK-3β acts as a negative regulator of GLI1, most likely via its phosphorylation and subsequent proteasomal degradation [71]. However, to our knowledge, it has never been demonstrated that GSK-3β phosphorylates GLI1. Whatever the case, perifosine, by inhibiting Akt, induced a decrease in Ser9-phosphorylated (hence inactive [219]) GSK-3β in T-ALL cell lines. Consistently, perifosine synergized with GANT61 in decreasing T-ALL cell viability [212]. Regarding the role of ERK in HH signaling, it is known that ERK2 phosphorylates GLI1 at multiple sites (Ser102, Ser116, and Ser130) which are located near the high-affinity binding site for SUFU. The ERK2-mediated phosphorylation weakens the binding of GLI1 to SUFU, thereby increasing GLI1 transcriptional activity [70] (Figure 4).

The HH pathway also crosstalks with Forkhead box C1 (FOXC1), a transcription factor which plays a pivotal role in mesenchymal lineage specification, normal organ development, and longevity [220]. However, FOXC1 has been demonstrated to drive cancer metastasis and to be an independent determinant of increased metastatic risk at the time of the initial diagnosis [221]. A long run (1–2 months) of GLI1 knockdown CUTLL1 T-ALL cells yielded rare GLI1-deficient populations which showed compensatory upregulation of GLI2 due to increased FOXC1 activity [222]. Mechanistically, in PTEN wild-type CUTLL1 cells, FOXC1 inhibited the ubiquitination of GLI2, possibly via an Akt-dependent mechanism which could involve downregulation of GSK-3β activity, as GSK-3β phosphorylates GLI2 on multiple Ser residues and targets it for proteasomal degradation [223] (Figure 5). However, in the case of PTEN null GLI1-deficient Jurkat T-ALL cells, the augmented expression of GLI2 was not due to increased Akt signaling but was, rather, associated with a decreased expression of histone acetyltransferase p300 (EP300). This observation suggests that reduced acetylation of GLI2 increases its stability, as we have seen with GLI1. Moreover, it is known that the N-terminal domain of FOXC1 binds directly to GLI2, thereby enhancing its DNA-binding and transcription-activating ability [224] (Figure 5). Whatever the case, the GLI2 overexpressing leukemic subsets displayed an extremely aggressive behavior in vivo, characterized by a massive meningeal dissemination, even in the absence of a BM infiltration [222]. Interestingly, a GLI2-orientated HH signaling activation signature is predictive of a worse prognosis, which suggests that low GLI1 levels may paradoxically make T-ALL patients highly susceptible to disseminated meningeal leukemia. This hypothesis is supported by the observation that patients with an extensive leukemic dissemination in the central nervous system display low GLI1 transcript levels [222]. Overall, these findings indicate that an SMO-independent, GLI2-dependent HH signaling activation may be implicated in meningeal infiltration in T-ALL patients who display persistently low levels of GLI1 expression. However, it remains to be determined how FOXC1 is upregulated when GLI1 levels remain low for an extended period of time. 

NOTCH signaling, which is aberrantly upregulated in >65% of T-ALL patients by activating mutations in NOTCH1, is a major regulator of leukemic cell growth, survival, and rewired metabolism. The oncogenic activity of NOTCH1 is critically dependent on the enhanced expression of c-MYC [225]. Therapeutic targeting of the NOTCH pathway with pan-NOTCH inhibitors (γ-secretase inhibitors or GSIs) has proven challenging in T-ALL patients, due to off-target toxicities and primary drug resistance [226,227]. There is evidence for a crosstalk between the NOTCH and HH pathways, as cyclopamine decreased the expression of both the cleaved, active fragment of NOTCH (intracellular NOTCH (ICN1)) and full-length NOTCH in NOTCH-mutated DND-41 T-ALL cells [203]. A combination treatment consisting of GSI and cyclopamine further downregulated the ICN expression. However, the combination displayed only an additive effect on the growth of the leukemic cells, as no synergism was detected. [203]. The interest in these findings is limited by the use of only one cell line, the lack of a suitable control (T-ALL cells harboring no NOTCH mutations) and primary samples, as well as the lack of experiments performed in vivo in an animal model. A more recent study demonstrated a marked overexpression of GLI1, PTCH1, PTCH2, and SMO in ICN1-transformed murine T-ALL cells compared to their healthy counterpart, i.e., thymic DP cells. Moreover, GANT61 was highly cytotoxic in mouse NOTCH-dependent T-ALLs [222]. However, human T-ALL cell lines and PDXs expressing wild type NOTCH were significantly more sensitive to treatment with GANT 61 compared to NOTCH mutant cells. Interestingly, treatment of NOTCH mutated T-ALL cells with a GSI significantly sensitized leukemic cells to GANT61, whereas this effect was not observed in a NOTCH1 wild-type T-ALL cell line (UPALL-13) [222]. Therefore, the real impact of NOTCH1 mutations on HH signaling in human T-ALL cells is still undefined.

The interactions between the glucocorticoid receptor nuclear receptor subfamily 3 group C member 1 (NR3C1) and the HH pathway [228] are of particular interest given that glucocorticoids (GCs) are a cornerstone of T-ALL therapy and a poor response to prednisone is an early marker of unfavorable outcome in T-ALL patients [229]. Mechanistically, dexamethasone (DEX) reduces GLI1 transcriptional activity (as demonstrated by PTCH1 and HHIP downregulation) while initially upregulating its expression levels in HSB2 and DND-41 T-ALL cells. GLI1 and NR3C1 physically interact in DND-41 cells; however, the degree of acetylation of NR3C1 is enhanced by DEX [228]. This observation is intriguing in light of the known inhibitory effects of acetylation on the transcriptional activity of both GLI1 and GLI2 via the occupancy of their target promoter [77,230]. Moreover, it is well established that activated NR3C1 interacts with both histone acetyltransferases (HATs) and histone deacetylases (HDACs) at promoter sites [231]. Accordingly, DEX treatment causes a reduction in HDAC1 which immunoprecipitates with either GLI1 or NR3C1, whereas immunoprecipitated HAT p300/CBP-associated factor (PCAF) increases.

Furthermore, DEX treatment reduces GLI1 stability in T-ALL cells via enhanced ubiquitination and proteasomal degradation [228] (Figure 6). Overall, the results of this study suggest that when DEX activates NR3C1, there is initially an accumulation of acetylated (hence inactive) GLI1, which is then ubiquitinated and proteolytically degraded. Therefore, the crosstalk between GLI1 and NR3C1 signaling pathways might be exploited in T-ALL patients by increasing the therapeutic efficacy of GCs. Interestingly, ATO at low doses sensitizes GC-resistant T-ALL cells to dexamethasone via an Akt-dependent pathway [232]. However, given the crosstalk between HH signaling and NR3C1, it could not be ruled out that part of the effects of APL on GC-resistance are related to inhibition of the HH pathway. 

Another pathway known for its interactions with HH signaling is the AMP-activated protein kinase (AMPK) network. For instance, AMPK is capable of directly phosphorylating and destabilizing GLI1 protein, thereby blocking GLI1 transcriptional activity in medulloblastoma [233]. AMPK-dependent phosphorylation at Ser 102, Ser408, and Thr1074 promotes retention of GLI1 in the cytoplasm by opposing its nuclear translocation. Moreover, the phosphorylation results in GLI1 degradation via β-transducin repeat containing E3 ubiquitin protein ligase (β-TrCP)-mediated ubiquitination [234]. AMPK is a master regulator of cell metabolism and confers neoplastic cells the ability to endure metabolic stresses [235]. The AMPK pathway is active in T-ALL cells [236], where it balances glycolysis and mitochondrial metabolism for controlling leukemic stress and survival [237]. Knockout of AMPK catalytic subunit (AMPKα1) by CRISPR–Cas9 technology increased (≈1.5–2 fold) GLI1 protein levels in T-ALL cells. Interestingly, PDX samples from T-ALL patients expressing higher levels of GLI1 were inclined to display lower levels of active (i.e., phosphorylated at Thr172) AMPK. In contrast, PDX specimens harboring higher levels of active AMPK had the tendency to show lower levels of GLI1. This inverse relationship between AMPK and GL1 was more evident in PDX samples displaying wild-type NOTCH1 and expressing PTEN [238]. Increased sensitivity to GANT61 was detected following either genetic inactivation of AMPKα1 or pharmacological inhibition of AMPK by Compound C. Moreover, combined targeting of HH and AMPK signaling pathways in T-ALL cells by GANT61 and SBI-0206965 (another AMPK inhibitor) significantly increased the cytotoxic effects of the drugs [238].

In T-ALL cells, AMPK signaling was the most consistently downregulated pathway under serum-depleted conditions, and this coincided with increased GLI1 expression and enhanced sensitivity to HH inhibitors, especially GANT 61. In contrast, mTORC1/p70SK1 activity was upregulated. Serum starvation in vitro mimics, at least in part, the growth of metabolically stressed cancer cells trying to adapt in vivo to nutrient-depleted environments, such as BM and the thymus [238]. The increase in GLI1 protein evoked by serum deprivation may be related to both increased stability of the transcription factor, caused by enhanced p70S6K1-dependent Ser84 phosphorylation [72], and decreased proteolytic degradation [233,239] (Figure 7). These findings suggest that combined therapeutic targeting of AMPK and HH signaling pathways may represent a strategy in rapidly growing T-ALL cases where nutrient availability is limited.

#### 7.1.4. Regulation through Non-Coding RNAs

For the sake of completeness, it should be recalled that in T-lymphoblastic lymphoma (T-LBL), HH activation is dependent on the downregulation of some miRNAs (miR30a, miR-141, miR-193b) which positively impact on the transcription of SMO [240]. The distinction between T-ALL and T-lymphoblastic lymphoma (T-LBL) depends on the level of BM involvement, with T-ALL cases defined by having 20% or more blasts in the BM, whereas T-LBL cases have less than 20% BM blasts with predominance of an extramedullary disease [241]. However, the two entities are closely related [242,243]. Therefore, it could not be ruled out that in some T-ALL cases, HH signaling dysregulation is due to miRNA alterations, which have been implicated in childhood T-ALL pathogenesis [244]. Indeed, non-coding RNas, including miRNAs and long non-coding RNAs (lncRNAs) are rapidly emerging as key regulators of aberrant HH signaling in cancer cells [245,246,247]. 

### 7.2. Oncogenic Potential of HH Mutations in T-ALL

T-ALL is characterized by the transformation of immature T-cells, which accumulate a wide array of genetic and epigenetic anomalies. These alterations lead to the sustained proliferation and survival of abnormal T-cells [248]. For example, dysregulation of NOTCH1 signaling due to NOTCH1 mutations leads to T-ALL in both mice and humans [249,250]. The evidence accumulated so far suggests that an aberrant activation of the HH pathway is not per se leukemogenic in mice. For instance, when the 2 SMO mutations identified by Dagklis and coworkers [9] were transduced in murine hematopoietic progenitor cells, 11 of 13 mice expressing the mutants showed enlarged lymph nodes infiltrated by cells. However, none of the mice developed acute leukemia, thereby implying that activation of the HH pathway is not sufficient to cause T-ALL, although it could block T-cell development in vivo. These findings were confirmed by the the same group via the ectopic expression of either SHH or IHH together with a JAK3(M511I) mutant in BM cells of Balb/C mice. The JAK3 mutant causes the development of a long latency T-ALL, characterized by the accumulation of immature CD8^+^ leukemic T-cells [251]. Interestingly, there was a strong correlation between GLI1 and JAK3 expression in human T-ALL specimens [10], which suggests that activation of the HH pathway could somehow synergize with activated JAK3 signaling in T-ALL development. Coexpression of either SHH or IHH with the mutant JAK3 in Balb/C mice does not significantly shorten the latency of the disease. However, the leukemic cells expressing both the HH ligand and mutant JAK3 became the dominant clone in all organs. The mice overexpressing the HH ligand and the JAK3 mutant also displayed more infiltration of neoplastic cells in lymph nodes and a lower white blood cell count than those expressing the JAK3 mutant alone. These findings suggest that the leukemic clones overexpressing both the HH ligand and the JAK3 mutant have a clonal advantage over the clones expressing either the HH ligand or the JAK3 mutant alone. Moreover, there was an increased homing of the cancer cells to hematopoietic organs [10]. Yet another interesting finding which emerged from this study is that TECs from animals overexpressing either SHH or IHH in their T-cells displayed an increased activation of the HH pathway as compared with the TECs isolated from wild-type mice. Furthermore, TECs showed augmented expression of δ-like canonical NOTCH ligand 4 (DLL4), interleukin-7 (IL-7), and vascular endothelial growth factor (VEGF). All of these ligands affect the development of T-cells and stimulate their survival and proliferation [252]. These observations clearly indicate that SHH or IHH, when expressed by T-ALL cells, somehow affect normal TECs in the leukemic microenvironment, thereby stimulating them to produce more ligands necessary for the survival and proliferation of the developing leukemic cells.

Another clear indication that the activation of the HH pathway contributes to the survival and maintenance of T-ALL cells comes from a study performed in a zebrafish model, where notch1a promotes T-cell transformation [253]. Burns and coworkers [206] selectively expressed in immature zebrafish T-cell progenitors a CRISPR/Cas9-generated mutant ptch1 known to induce an aberrant upregulation of the HH pathway [254]. The expression of the mutant ptch1 leads to a significant acceleration of the onset and to an increased penetrance of notch1-induced T-ALL. The median time to T-ALL onset was 13 weeks in the ptch1 fish, versus 39 weeks in the controls [206]. Overall, these data demonstrate that the activation of HH signals through the disruption of its negative regulator ptch1 acts as an oncogenic driver in zebrafish T-ALL. 

### 7.3. Therapeutic Targeting of HH Signaling in T-ALL

Preclinical studies performed in vitro or in vivo have demonstrated the efficacy of HH signaling inhibitors in T-ALL cells. Hou and coworkers [212] reported that cyclopamine was effective in reducing the viability of human T-ALL cell lines when employed at a concentration (10 μM) which is known for inducing off-target effects [63]. The IC_50_ of cyclopamine is around 1 μM [255]. However, when cyclopamine was used at 1–3 μM, its efficacy was extremely modest in T-ALL cell lines. In contrast, the GLI inhibitor GANT 58 was more cytotoxic than cyclopamine. A drug combination consisting of GANT 58 and perifosine (or GSK690693, a different Akt inhibitor) displayed synergistic cytotoxic effects in T-ALL cell lines [212]. These findings indicate that combined treatments based on inhibition of PI3K/Akt and GLI1 could result in an improved efficacy in targeting T-ALL cells. 

In a subsequent study, two SMO inhibitors (cyclopamine and GDC-0449) caused a concentration-dependent inhibition of proliferation in three of nine T-ALL cell lines, while GANT61 inhibited proliferation in six of nine cell lines. Importantly, cell lines in which the drugs led to the strongest decrease in GLI1 expression also displayed the strongest inhibition of proliferation [10]. Moreover, ex vivo incubation of PDX T-ALL samples showing GLI1 expression with either GDC-0449 or GANT61 led to a significant decrease in proliferation. In contrast, PDX samples with low or undetectable GLI1 expression levels were insensitive to HH pathway inhibitors. Similar results were obtained with xenografts into immune-deficient NSG mice which were treated with GDC-0449 or GANT 61 [10]. Overall, these findings demonstrate the specificity of the drugs used in this report. Moreover, the same study showed that T-ALL cell lines which respond to GANT 61 were sensitized to ARA-C or doxorubicin cytotoxic effects in the presence of GANT 61. In contrast, in T-ALL cells which were insensitive to GANT 61, a combined GANT 61/ARA-C treatment was not more cytotoxic than ARA-C alone [10]. These data are in agreement with the observation that HH mutations are associated with resistance to the initial induction cycle of intensive polychemotherapy [206]. Drug resistance may be partly related to the fact that HH signaling upregulation drives glucuronidation of nucleoside analogs, thereby lowering their cytotoxic effects [256]. Moreover, activation of HH signaling is associated with GC-resistance in CEM T-ALL cells [257].

Overall, the studies discussed in this Section and in Section 7.1.3 provide evidence that a subgroup of T-ALL patients, displaying activation of the HH signaling pathway, may benefit from treatment with SMO or GLI1 inhibitors, alone or in combination with chemotherapy or other targeted therapeutics.

## 8. Perspectives and Conclusions

In the past several years, the outcomes of T-ALL patients (especially children) have dramatically improved, despite the fact that the mainstays of therapy have not changed. Nevertheless, therapy treatment-related deaths and long-term detrimental adverse effects for leukemia survivors remain serious issues which still await a solution. Moreover, the prognosis for most refractory and relapsed patients remains dismal due to the few available treatment options. Therefore, there is an urgent need for novel and less toxic targeted therapeutics for T-ALL patients. The evidence we have reviewed in this article indicates that targeting the HH signaling pathway may present a new therapeutic opportunity in patients who are resistant to conventional drugs such as ARA-C or GCs. Adapting T-ALL therapy to patients’ individual needs requires a growing arsenal of small molecule inhibitors and chemotherapeutic agents for the simultaneous targeting of multiple tumor-promoting processes and oncogenic signaling pathways. As we have summarized in this review, we now have at our disposal several drugs targeting HH signals. While acknowledging the relevance of these therapeutics, much is left to be uncovered for translating them in the clinic for treatment of T-ALL patients. Moreover, although some of these drugs have been approved for cancer treatment, there is a need for new therapeutics to overcome drug resistance and adverse effects of the inhibitors currently used. In particular, given the modest activity of the SMO inhibitors in human T-ALL cells and the well-known issue of emergence of resistance to these therapeutics in other tumor clinical trials, it will be crucial to determine which signaling pathways alter leukemic cell sensitivity to SMO inhibitors. These findings will be of the utmost importance to design rational combination treatments incorporating HH pathway inhibitors in T-ALL patients.

We need to achieve a much better understanding of the complex molecular interactions involving HH signaling and other pathways in both healthy and neoplastic cells as well as the best ways to employ drugs targeting HH in T-ALL. As we learn more about the roles of HH in T-ALL, it may be possible to develop drugs which mainly target the functions of HH in leukemic cells and which could be more amenable to a therapeutic intervention than HH signaling itself. For example, a recent study performed in melanoma cells has revealed that a novel SOX2-BRD4 transcriptional complex drives the expression of GLI1 via non-canonical HH signaling [258]. Interestingly, combined targeting of the canonical HH pathway with a SMO inhibitor MRT-92 and of the SOX2-BRD4 complex using a potent proteolysis targeted chimeras (PROTACs)-derived BRD4 degrader, yielded a synergistic anti-proliferative effect in melanoma cells both in vitro and in vivo. Moreover, the indirect inhibition of GLI1 activity by affecting its acetylation status (for example by HDAC inhibitors) could represent an interesting alternative therapeutic approach when combined with GC, as we have highlighted in this manuscript. This strategy warrants further clinical evaluation.

Furthermore, biomarkers indicating which patients may benefit the most from drugs targeting HH are still awaiting definitive identification, although some progress has been made, as we have summarized above. 

In conclusion, despite some significant issues which remain to be solved, the findings we have summarized here provide a compelling rationale for future clinical trials aimed at testing the potential of HH inhibitors in T-ALL patients, especially in combination with either chemotherapeutics or other targeted drugs (e.g., MEK/ERK inhibitors, PI3K/Akt/mTOR inhibitors, AMPK inhibitors, and others). 

## Figures and Tables

**Figure 1 ijms-24-02962-f001:**
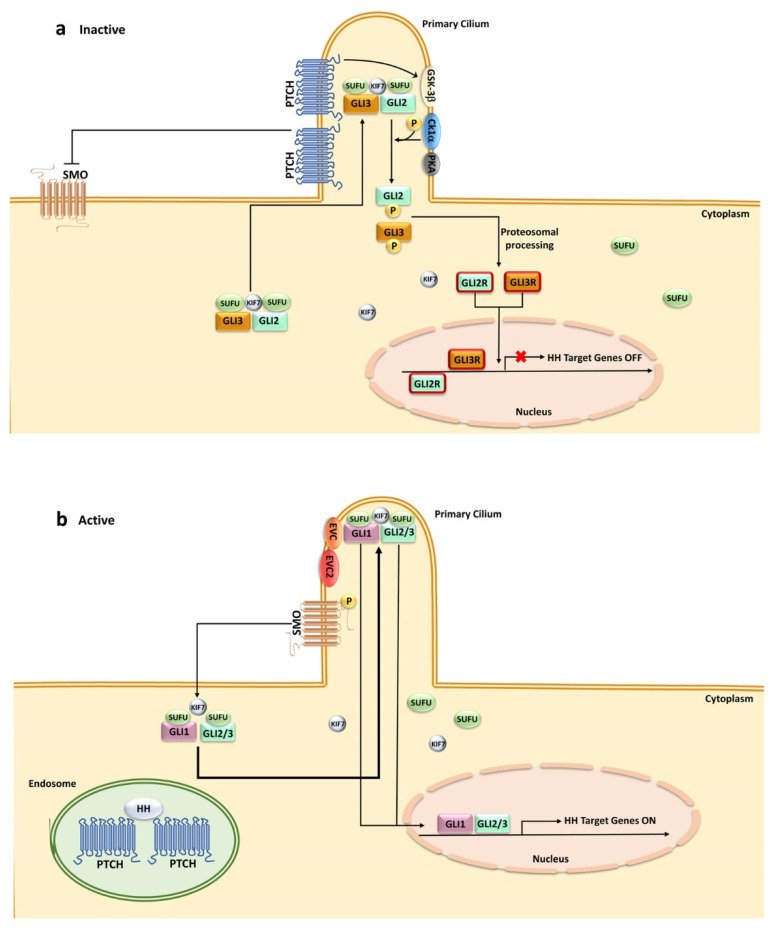
HH signaling in mammals. A simplified cartoon illustrating activation of HH pathway in mammalian cells: (**a**) in the absence of HH ligands, PTCH is localized at the primary cilium while SMO remains distant from the cilium. PTCH inhibits SMO by preventing its cilial entry. Several kinases (GSK-3β, CK1-α, PKA) are recruited by the GLI/SUFU/KIF7 complexes and activated by PTCH. GLI transcription factors are phosphorylated by these kinases, which promote their processing into the repressor forms, translocating to the nucleus, and thereby blocking HH target gene transcription; (**b**) HH ligand binding to PTCH relieves SMO inhibition. PTCH is endocytosed and displaced from the cilium, whereas SMO accumulates in the cilium in an active state. The GLI/SUFU/KIF7 complexes are recruited by SMO. SMO activation results in the detachment of GLI transcription factors from SUFU and KIF7 with the aid of EvC and EvC2. GLI transcription factors migrate to the nucleus, activating gene transcription. CK1-α, casein kinase 1-α; EvC, Ellis–Van Creveld Syndrome; GLI-R, GLI repressor; GSK-3β, glycogen synthase kinase 3β; KIF7, kinesin family protein 7; PKA, protein kinase A; PTCH, Patched; SMO, Smoothened; SUFU, suppressor of fused.

**Figure 2 ijms-24-02962-f002:**
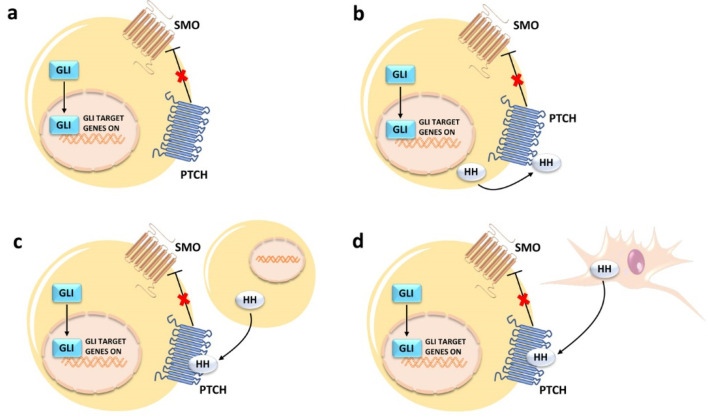
Different modalities of canonical HH signaling activation in cancer cells: (**a**) type I ligand-independent; (**b**) type II ligand-dependent autocrine; (**c**) type IIIa ligand-dependent; (**d**) type IIIb ligand-dependent reverse paracrine. For details, see the text. HH, hedgehog ligand.

**Figure 3 ijms-24-02962-f003:**
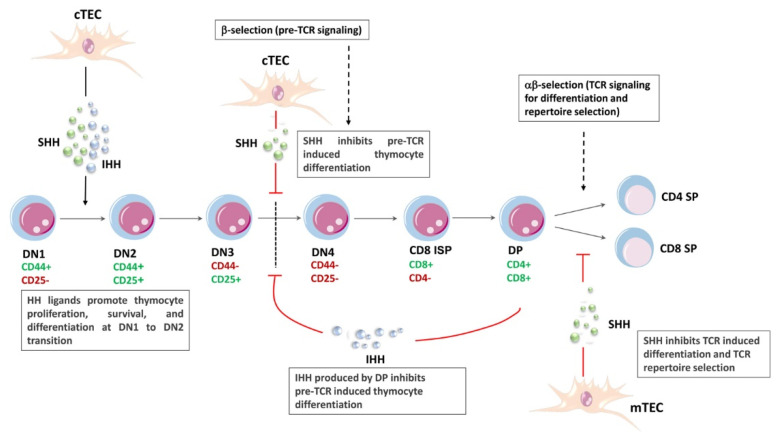
Multiple stages of thymocyte differentiation are controlled by HH signaling. The schematic cartoon illustrates how HH signaling influences several stages of thymocyte development in the mouse embryonic thymus. cTECs, cortical thymic epithelial cells; IHH, Indian hedgehog; ISP, immature single positive; mTECs, medullary thymic epithelial cells; SHH, Sonic hedgehog; TCR, T-cell receptor.

**Figure 4 ijms-24-02962-f004:**
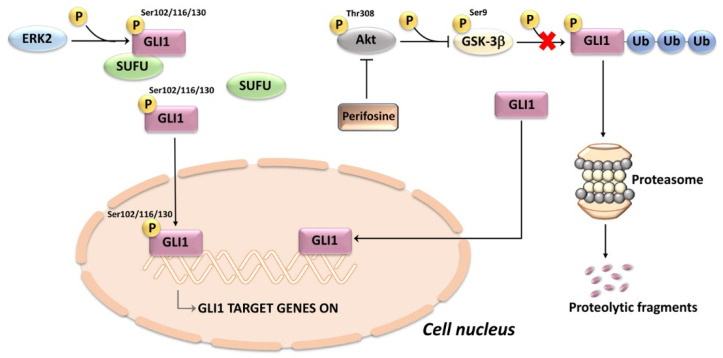
ERK2 and Akt upregulate the transcriptional activity of GLI1 in T-ALL cells. The schematic cartoon illustrates possible mechanisms of GLI1 activation through ERK2 and Akt signaling. ERK2 phosphorylates GLI1 at Ser residues, thereby loosening the interactions between SUFU and GLI1 which is free to translocate to the nucleus. Active (Thr 308 phosphorylated) Akt phosphorylates GSK-3β at Ser 9, thereby inhibiting GSK-3β which cannot target GLI1 for ubiquitination and proteasomal degradation. The Akt inhibitor perifosine restores GLI1 proteolysis. GSK-3β, glycogen synthase kinase 3β; SUFU, suppressor of fused.

**Figure 5 ijms-24-02962-f005:**
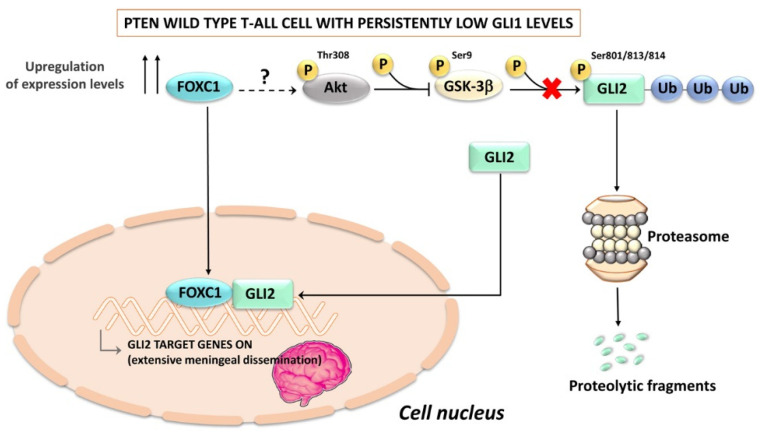
Persistently low GLI levels result in upregulation of GLI2 via FOXC1 in PTEN wild-type T-ALL cells. The schematic cartoon illustrates a possible mechanism of GLI2 upregulation via FOXC1/Akt/GSK-3β signaling. Upregulated FOXC1 activates Akt through a still undefined mechanism. Active Akt phosphorylates GSK-3β at Ser 9, thereby inhibiting GSK-3β which cannot target GLI2 for ubiquitination and proteasomal degradation. GLI2 translocates to the nucleus, where it upregulates a specific transcriptional program, resulting in extensive leukemic meningeal dissemination. FOXC1; Forkhead box C1; GSK-3β, glycogen synthase kinase 3β.

**Figure 6 ijms-24-02962-f006:**
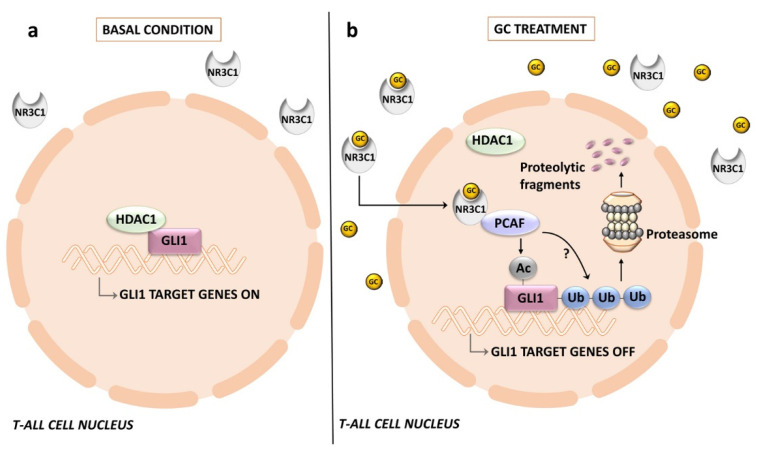
Crosstalk between HH signaling and the glucocorticoid receptor NR3C1 in T-ALL cells: (**a**) in the absence of glucocorticoids, GLI1 is associated with HDAC1 and transcription of GLI1-regulated genes is active; (**b**) upon treatment with glucocorticoids, NR3C1 translocates to the nucleus, thereby displacing HDAC1 and facilitating association of the histone acetylase PCAF with GLI1. PCAF acetylates GLI1, which is then targeted for proteasomal degradation. GC, glucocorticoids; HDAC1, histone deacetylase 1; NR3C1, nuclear receptor subfamily 3 group C member 1; PCAF, p300/CBP-associated factor.

**Figure 7 ijms-24-02962-f007:**
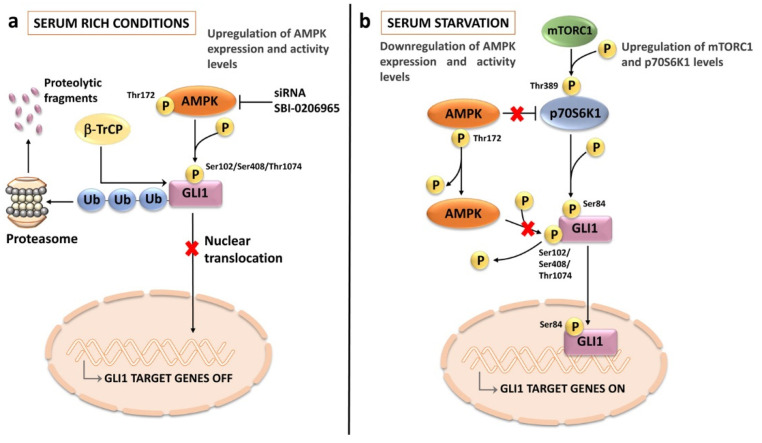
Interactions between AMPK and HH signaling pathways in T-ALL cells: (**a**) under serum-rich conditions, AMPK is overexpressed and active (i.e., phosphorylated at Thr172). Active AMPK phosphorylates GLI1 at multiple amino acid residues. This phosphorylation leads to GLI1 ubiquitination by β-TrCP followed by proteasomal degradation; (**b**) when T-ALL cells are serum starved, AMPK expression and activity levels are downregulated, hence GLI1 is not phosphorylated at AMPK-dependent residues. In contrast, the mTORC1/p70S6K1 axis is overactive, due to AMPK inactivation/decreased expression. p70S6K1 kinase phosphorylates GLI1 at Ser84, thereby increasing the stability of GLI1 and enhancing its transcriptional activity. AMPK, AMP-activated protein kinase; β-TrCP, β-transducin repeat containing E3 ubiquitin protein ligase.

**Table 1 ijms-24-02962-t001:** SMO inhibitors used in clinical trials.

SMO Inhibitors	National Clinical Trial (NCT)	Tumor Type
Vismodegib (GDC-0449)	e.g., NCT03035188, NCT02115828	BCC, prostate cancer
Sonidegib (NVP-LDE225)	e.g., NCT02111187, NCT04066504,	Prostate cancer, BCC,
Glasdegib (PF-04449913)	e.g., NCT03466450, NCT04231851	Glioblastoma, AML
Saridegib (IPI-926)	e.g., NCT01609179	BCC
Taladegib (LY-2940680)	e.g., NCT01226485	Advanced cancer
TAK-441	e.g., NCT01204073	Advanced nonhematologic malignancies
BMS-833923 (XL139)	e.g., NCT01413906, NCT00670189	Solid tumors, BCC
Itraconazole	e.g., NCT01791894	BCC

## Data Availability

Data sharing not applicable. No new data were created or analyzed in this work.

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
