# Peer review of "Understanding the Roles of the Hedgehog Signaling Pathway during T-Cell Lymphopoiesis and in T-Cell Acute Lymphoblastic Leukemia (T-ALL)"

_ijms, 2023, doi:10.3390/ijms24032962_

Round 1
Reviewer 1 Report
The purpose of the manuscript is to provide a comprehensive, review oft he role of Hedghog signaling pathway in T-cell lymphopoiesis and T-ALL. This is a review by a group of authors who are the experts in T-cell acute lymphoblastic leukemia (T-ALL). The authors provide a comprehensive, updated review of the role of the Hedgehog (HH) signaling pathway in regulation of gene expression and cellular proliferation in T-ALL.
The hyper-activation of HH signaling has been demonstrated in various types of malignant diseases, including medulloblastoma, rhabdomyosarcoma, glioblastoma, and cancers of colon, lung, pancreas stomach, ovary, and breast. Here, the authors provide a comprehensive review and mechanistic rationale for the role of HH signaling in T-ALL. The role of HH signaling network in regulation of downstream pathways, as well as gene expression in T-ALL is reviewed extensively.
This is an outstanding review by leaders in the signal transduction field and T-ALL, that covers in detail an important topic that is of general interest for a broad scientific audience. The comprehensive approach that analyzes the effect of HH signaling network on T-cell lymphopoiesis and T-ALL provides a complete picture of this important topic. The strong translational component and review of the use of HH signaling pathway inhibitor for potential treatment is a strength of this paper.
The topic is very relevant in this field and addresses a specific gap in the field. There are very few review papers that address this topic. The paper adds a translational component which distinguishes it from other review manuscripts. The references are appropriate. The figures are outstanding and help the broad readership to understand the manuscript.
In summary, this is an excellent manuscript that includes a review of the basic mechanisms of HH signaling pathway in T-cell lymphopoiesis and T-ALL, along with a review of targeted therapeutic approaches. The review covers an important topic and the strong translational component of the paper makes this manuscript particularly suitable for publication in IJMS.
Author Response
Reviewer 1
Our answer: We thank the reviewer very much for the positive comments about our work.
Reviewer 2 Report
The manuscript presents the discussion of Hedgehog signaling pathway roles in vitro and in vivo at different diseases. The topic of review is new and promising. The review gives the detailed analysis of Hedgehog signaling pathway roles based on 243 references which give a deep insight in the topic discussed. The review is excellently illustrated by 7 figures. The authors critically analyzed modern knowledge and combined an unique point of view on the problem. Review is easy to read and understand. The discussion is very good and the conclusions are supported by the data. As for English, I am not native speaker, for me English is perfect. I did not find any serious mistakes to mention. It's a rare situation when I recommend to publish it as it is.
Author Response
Reviewer 2
Our answer: We thank the reviewer very much for the very positive comments about our work.
Reviewer 3 Report
A highly controlled, synchronized network of extracellular ligands, membrane receptors, signaling molecules, co-regulators, and transcription factors makes up the HH signaling pathway. This signaling cascade's interconnections are spatially and temporally controlled, so that pathway activation only occurs in the appropriate cell and tissue context. Therefore, both tissue growth and homeostasis depend on HH signaling activation. The PTCH and SMO proteins act as the HH pathway's gatekeepers. Dysregulation of the HH pathways is thought to emerge in various cancers, including hematological malignancies. HH has been reported active in lymphoid malignancies, including T- cell acute lymphoblastic leukemia (T-ALL). In this article, the authors discussed HH signaling in healthy and cancer cells and its relevance during T-cell lymphopoiesis and T-ALL. They also summarized a novel strategy targeting the HH signaling network for possible innovative therapies for a malignant blood disorder. Figures and discussion on T- cell lymphopoiesis is the main strength of this article. The article is well written. However, I have a suggestion for this article.
The authors should cut the general overview content from sections 2 and 3 and emphasize T-cell lymphopoiesis and T-ALL instead. A number of studies have covered the topics of canonical and noncanonical HH signaling.
Abstract: I advise the authors to reduce the length of the abstract. For example, Glasdegib, has recently been approved------induction chemotherapy" can be removed from the abstract. Using this statement unnecessarily dilutes the text.
Section 4. Therapeutic targeting of HH in cancer cells. When the authors mention targeting HH in cancer cells, they should include some nano drug therapy as well? Recent studies demonstrate that nanoparticles target the HH signaling pathway specifically. One such is the PBM nanoparticles' selective targeting of the hedgehog signaling system. PMID: 32867229
In addition, I advise the authors to provide a table for the inhibitors that lists clinical studies.
Author Response
Reviewer 3
The authors should cut the general overview content from sections 2 and 3 and emphasize T-cell lymphopoiesis and T-ALL instead. A number of studies have covered the topics of canonical and noncanonical HH signaling.
Our answer: we have cut the content from sections 2 and 3. We have expanded the Section on T-cell lymphopoiesis (see lines 346-357; and lines 372-379) as well as the Section on T-ALL (see lines 49-62; lines 453-480; and lines 870-875).
Abstract: I advise the authors to reduce the length of the abstract. For example, Glasdegib, has recently been approved------induction chemotherapy" can be removed from the abstract. Using this statement unnecessarily dilutes the text.
Our answer: that sentence has been deleted (see Abstract: lines 25-26). In general, the length of the abstract has been reduced from 199 to 158 words.
Section 4. Therapeutic targeting of HH in cancer cells. When the authors mention targeting HH in cancer cells, they should include some nano drug therapy as well? Recent studies demonstrate that nanoparticles target the HH signaling pathway specifically. One such is the PBM nanoparticles' selective targeting of the hedgehog signaling system. PMID: 32867229
Our answer: we have added a new subsection on nanoparticles (see lines 293-307).
In addition, I advise the authors to provide a table for the inhibitors that lists clinical studies.
Our answer: we have added the Table, as requested (see page 7).
Reviewer 4 Report
International Journal of Molecular Sciences (Manuscript ID: ijms-2139204), Comments to the Authors:
Title: Understanding the roles of the Hedgehog signaling pathway during T-cell lymphopoiesis and in T-ALL
Comments
The submitted review summarized recent findings on the biological roles and pathophysiology of the Hedgehog (HH) pathway during normal T-cell lymphopoiesis and in T-ALL. The authors discussed the potential therapeutic strategies that might expand the clinical usefulness of drugs targeting the HH pathway in T-ALL.
I think the submitted review can be accepted for publication after the authors respond to the following comments:
1. The authors should compare their review with previous similar reviews that discussed similar topic and were not cited in the current review including: A) Bongiovanni, Deborah, Valentina Saccomani, and Erich Piovan. 2017. "Aberrant Signaling Pathways in T-Cell Acute Lymphoblastic Leukemia" International Journal of Molecular Sciences 18, no. 9: 1904. https://doi.org/10.3390/ijms18091904; B) Crompton, T., Outram, S. & Hager-Theodorides, A. Sonic hedgehog signalling in T-cell development and activation. Nat Rev Immunol 7, 726–735 (2007). https://doi.org/10.1038/nri2151; C) Smelkinson, Margery G. 2017. "The Hedgehog Signaling Pathway Emerges as a Pathogenic Target" Journal of Developmental Biology 5, no. 4: 14. https://doi.org/10.3390/jdb5040014.
2. The authors should discuss the rationale behind writing this review. Why did they write this review?
3. What is the authors' contribution to the field of the Hedgehog signaling pathway?
4. The title should not contain any abbreviation. The authors should spell out “T-ALL”.
Author Response
Reviewer 4
- The authors should compare their review with previous similar reviews that discussed similar topic and were not cited in the current review including: A) Bongiovanni, Deborah, Valentina Saccomani, and Erich Piovan. 2017. "Aberrant Signaling Pathways in T-Cell Acute Lymphoblastic Leukemia" International Journal of Molecular Sciences 18, no. 9: 1904. https://doi.org/10.3390/ijms18091904; B) Crompton, T., Outram, S. & Hager-Theodorides, A. Sonic hedgehog signalling in T-cell development and activation. Nat Rev Immunol 7, 726–735 (2007). https://doi.org/10.1038/nri2151; C) Smelkinson, Margery G. 2017. "The Hedgehog Signaling Pathway Emerges as a Pathogenic Target" Journal of Developmental Biology 5, no. 4: 14. https://doi.org/10.3390/jdb5040014.
Our answer: we now mention the review by Bongiovanni et al. (ref. 9) and Crompton et al. (ref 130), which are both related to the topic of our manuscript. Our review is a summary of this research field current as of January 2023. We respectfully disagree with the reviewer on the review by Smelkinson, as this article deals with HH signaling involvement in pathogen infections. Therefore, we have not included it.
2.The authors should discuss the rationale behind writing this review. Why did they write this review?
Our answer: we aimed to write a comprehensive and updated review on HH signaling whose therapeutic importance is becoming more and more appreciated in T-ALL (see lines 49-62).
3.What is the authors' contribution to the field of the Hedgehog signaling pathway?
Our answer: A. Martelli and F. Chiarini are among the authors of an article on HH signaling in AML (see J Hematol Oncol . 2017 Jan 21;10(1):26. doi: 10.1186/s13045-017-0396-0.).
- A. McCubrey has published several articles on HH signaling
(see Adv Biol Regul.2017 Aug;65:59-76. doi: 10.1016/j.jbior.2017.06.002. Epub 2017 Jun 6.;
Oncotarget. 2017 Feb 21;8(8):14221-14250. doi: 10.18632/oncotarget.13991.;
Biochim Biophys Acta. 2016 Dec;1863(12):2942-2976.;
Adv Biol Regul. 2015 Sep;59:65-81. doi: 10.1016/j.jbior.2015.06.003. Epub 2015 Jul 17.;
Oncotarget. 2014 May 30;5(10):2881-911. doi: 10.18632/oncotarget.2037.).
4.The title should not contain any abbreviation. The authors should spell out “T-ALL”.
Our answer: the mistake has been corrected.
Reviewer 5 Report
In the submitted review manuscript Martelli et al. gave an overview of the role of Hedgehog-GLI signaling pathway in T-cell lymphopoiesis as well as in T-cell acute lymphoblastic leukemia. This manuscript is comprehensive and well written. As all HH-GLI pathway-related reviews, it covers comprehensive introduction which have very well known facts. Nevertheless, the main topic of this review is satisfactorily covered and well structured.
There are just few minor typos (line 273 "GANT 61" and lines 300 "BCC"), while symbols of human genes should be consistently written in italics, especially taking care when authors write about either gene or protein mutations.
Author Response
Reviewer 5
In the submitted review manuscript Martelli et al. gave an overview of the role of Hedgehog-GLI signaling pathway in T-cell lymphopoiesis as well as in T-cell acute lymphoblastic leukemia. This manuscript is comprehensive and well written. As all HH-GLI pathway-related reviews, it covers comprehensive introduction which have very well known facts. Nevertheless, the main topic of this review is satisfactorily covered and well structured. There are just few minor typos (line 273 "GANT 61" and lines 300 "BCC"), while symbols of human genes should be consistently written in italics, especially taking care when authors write about either gene or protein mutations.
Our answer: we thank the reviewer very much, those typos as well as the issue with italicized genes have been corrected.
Reviewer 6 Report
This is an excellent review that provides a comprehensive summary of this field. It will serve as a useful resource for those new to this area as well as researchers in this field. The coverage of the topic is very good, including both selection of references and the descriptions of findings from those references. Figures included in the manuscript are clearly presented and help to clarify and summarise the commentary in the text. English language is also excellent, with sentence and paragraph structure appropriate and easy to follow. Minor typographical / grammatical errors are listed along with some minor queries. I recommend publication of this manuscript pending correction of the minor errors below.
Minor corrections and questions:
Line 100: Should “gene expression” be “gene regulatory activities”?
Line 17: “Along with other…”
Line 123: “whereby” should be “thereby”
Line 155: Citation error (Yang, 1998 #281)
Line 203: “noncanonical HH signaling”
Line 207: “crosstalks” should be “crosstalk” (plural is the same as singular for this word)
Line 224: “whereby” should be “thereby”
Line 239: “academy” should be “academic”
Line 258: “about 8 months”
Line 311: “mouse models” (adjectives are always singular in English)
Line 318: “crosstalks” should be “crosstalk”
Line 329: “embryonic” (spelling error)
Line 373: “thymocyte” (singular when used as an adjective)
Line 412: “effects on thymocyte differentiation, similar to those” (spelling / grammatical errors)
Line 420: “FTOC experiments”
Line 420: “allowed to conclude” is not English. Suggested alternative “indicated”
Line 422: “children thymi” should be “thymi from children”
Line 496: “Interestingly”
Line 515: “inexistent” should be “nonexistent”
Line 530: “controls” should be “control”
Line 538: “ERK role” should be “the role of ERK”
Line 559: “targets if” should be “targets it”
Line 562: “expression in histone…” should be “expression of histone…”
Line 566: “overexpressing GLI2” should be “GLI2 overexpressing”
Line 616: “NR3C1 degree of acetylation” should be “degree of acetylation of NR3C1”
Line 637: “patients for increasing…” should be “patients by increasing…”
Line 665: “aminoacidic residues” should be “amino acid residues”
Line 675: “mimicks” should be “mimics”
Line 686: “transcription” (spelling error)
Line 711: “suggests that the activation of HH pathway” should be “suggests that activation of the HH pathway”
Lines 715-716: The meaning of this sentence is not clear, specifically “than with JAK3(M511I) alone”
Line 720: “interesting” (spelling error)
Line 787: “we have now” should be “we now have”
Line 810: “which still remain” should be “which remain” (tautology)
Author Response
Reviewer 6
This is an excellent review that provides a comprehensive summary of this field. It will serve as a useful resource for those new to this area as well as researchers in this field. The coverage of the topic is very good, including both selection of references and the descriptions of findings from those references. Figures included in the manuscript are clearly presented and help to clarify and summarise the commentary in the text. English language is also excellent, with sentence and paragraph structure appropriate and easy to follow. Minor typographical / grammatical errors are listed along with some minor queries. I recommend publication of this manuscript pending correction of the minor errors below (there is a list of typos).
Our answer: we thank the reviewer very much for the extremely valuable work he/she did. All the typos/grammatical errors have been corrected and the sentence on JAK3 has been rewritten (see lines 787-793).
Round 2
Reviewer 3 Report
All of the issues raised have been addressed by the authors. It might be approved for publishing.
Reviewer 4 Report
International Journal of Molecular Sciences (Manuscript ID: ijms-2139204), Comments to the Authors:
Title: Understanding the roles of the Hedgehog signaling pathway during T-cell lymphopoiesis and in T-ALL
Comments
After reading the authors’ response to my comments, I think the authors responded to my remarks and I believe the review can be accepted for publication.